# Application of Artificial Neural Networks to Predict the Catalytic Pyrolysis of HDPE Using Non-Isothermal TGA Data

**DOI:** 10.3390/polym12081813

**Published:** 2020-08-12

**Authors:** Mohammed Al-Yaari, Ibrahim Dubdub

**Affiliations:** Department of Chemical Engineering, King Faisal University, P.O. Box: 380, Al-Ahsa 31982, Saudi Arabia; malyaari@kfu.edu.sa

**Keywords:** high-density polyethylene (HDPE), catalytic pyrolysis, thermogravimetric analysis (TGA), activation energy, artificial neural network (ANN)

## Abstract

This paper presents a comprehensive kinetic study of the catalytic pyrolysis of high-density polyethylene (HDPE) utilizing thermogravimetric analysis (TGA) data. Nine runs with different catalyst (HZSM-5) to polymer mass ratios (0.5, 0.77, and 1.0) were performed at different heating rates (5, 10, and 15 K/min) under nitrogen over the temperature range 303–973 K. Thermograms showed clearly that there was only one main reaction region for the catalytic cracking of HDPE. In addition, while thermogravimetric analysis (TGA) data were shifted towards higher temperatures as the heating rate increased, they were shifted towards lower temperatures and polymer started to degrade at lower temperatures when the catalyst was used. Furthermore, the activation energy of the catalytic pyrolysis of HDPE was obtained using three isoconversional (model-free) models and two non-isoconversional (model-fitting) models. Moreover, a set of 900 input-output experimental TGA data has been predicted by a highly efficient developed artificial neural network (ANN) model. Results showed a very good agreement between the ANN-predicted and experimental values (R^2^ > 0.999). Besides, A highly-efficient performance of the developed model has been reported for new input data as well.

## 1. Introduction

Plastic wastes have become an irritating worldwide issue particularly in many developed countries, where a massive quantity is produced and disposed of. The main source of plastic wastes is municipal solid wastes (MSW) [1]. Globally, most of the plastic wastes are either disposed of in landfills or incinerated [2]. While landfill disposal is still considered as undesired and expensive treatment, the waste destruction by incineration is also expensive and has problems with high emissions and thus environmental concerns.

Combustion as a primary recycling technique is used to treat plastic wastes, but it is still restricted by environmental legislation. However, plastic wastes reshaping, as a secondary recycling method, is limited to only 20 wt % plastic wastes [3].

Recently, research effort is more focused on tertiary recycling, by involving some sophisticated technology such as pyrolysis, gasification, and catalytic cracking in the recycling industries [3]. Pyrolysis has some advantages over the rest of waste disposal techniques because of its low volume products (gases, liquids, and char) that can be used as fuel and can be added to petroleum refinery feedstocks or act as chemical feedstocks [4,5]. However, pyrolysis needs to be achieved at high temperature to have the desired type of oil. To overcome this challenge, catalytic cracking is used to reduce the cracking temperature [6,7].

Pyrolysis of high-density polyethylene (HDPE), representing about 17.8% of MSW plastic waste, has attracted the attention of some researchers. Kinetic parameters such as activation energy was targeted to be obtained for the design purposes of industrial processes. Conesa et al. [8] studied the HDPE pyrolysis using isothermal and non-isothermal thermogravimetric analysis (TGA) data at heating rates of 5, 25, 50, and 100 K/min. The reported activation energy values were ranging between 185 and 221.5 kJ/mol.

Aboulkas et al. [9] studied the thermal decomposition of HDPE by Friedman, Flynn–Wall–Qzawa (FWO), and Kissinger–Akahira–Sunose (KAS) iso-conversional methods at 2, 10, 20, and 50 K/min. Activation energy values within the range of 238–247 kJ/mol were reported.

Chin et al. [10] examined the pyrolysis of HDPE using TGA data at 10–50 K/min heating rates and 323–1173 K temperature range. Activation energies were found within 242.13–278.14 kJ/mol.

Diaz Silvarrey and Phan [11] developed a kinetic model for the TGA pyrolysis of HDPE. They applied Málek, KAS, and linear model fitting to calculate the kinetic parameters. All their experiments were performed under an N_2_ atmosphere at 5, 10, 20, and 40 K/min heating rates, and temperature range of 303–973 K. They reported different values of activation energy by different methods (202.40 ± 9.47 kJ/mol and 375.59 ± 39.69 kJ/mol by KAS and Friedman methods, respectively).

Khedri and Elyasi [12] obtained the kinetics parameters of the pyrolysis of HDPE using non-isothermal and isothermal TGA data by model-free models at 40, 45, 50, and 55 K/min heating rates. The calculated activation energies were reported to heavily vary with conversion and used methods.

This study aims to obtain activation energy of the catalytic pyrolysis of HDPE at different catalyst to polymer ratios and heating rates using non-isothermal TGA data. Three isoconversional methods and two non-isoconversional models have been used. Additionally, a highly-efficient artificial neural network (ANN) model has been developed, for the first time, to predict the pyrolytic behavior of the catalytic cracking of HDPE.

## 2. Materials and Methods

### 2.1. Experimental Procedure

Catalytic cracking of HDPE using the HZSM-5 catalyst has been investigated. Polymer samples were obtained from Ipoh SY Recycle Plastic, Malaysia. The proximate and ultimate analysis of HDPE were conducted using PerkinElmer Simultaneous Thermal Analyzer (STA) 6000, and PerkinElmer 2400 Series II CHNS Elemental Analyzer, Waltham, MA, USA, respectively. The characterization data are presented in Table 1. HZSM-5 was obtained from zeolite (CBV3024E) in ammonia form (Alfa Aesar, Ward Hill, MA, USA) and then converted into hydrogen form by calcination at 823 K and 2 K/min heating rate for 2 h in a muffle furnace. The catalyst specifications and the experimental matrix details are shown in Table 2 and Table 3, respectively. The test samples have been prepared with different catalyst to polymer mass ratios (0.5, 0.77, and 1.0). The cracking studies were performed using Mettler Toledo TGA/SDTA851^e^ (Polaris Parkway, Columbus, OH, USA) analyzer under 50 mL/min N_2_ as an inert gas. The results were evaluated with the V7.01 STAR^e^ software package. Heating rates of 5, 10, and 15 K/min were exerted. TGA equipment was used for the measurements in which the samples were heated from ambient temperature to 373 K for 5 min, and then heating continued to 523 K and was then held for 5 min. After that, heating continued to 973 K and the temperature was kept constant for another 5 min.

### 2.2. Kinetic Theory

For most kinetics, the rate of reaction (*r*) can be expressed as [13,14]:(1)$$r=\frac{d\alpha }{dt}=A\exp\left(-\frac{E}{RT}\;\right){\left(1-\alpha \right)}^{n}$$
where α is conversion, *t* is time, *A* is the pre-exponential factor, *E* is the activation energy, *R* is the universal gas constant, *T* is temperature, and *n* is the reaction order. Conversion can be calculated as follows:(2)$$\alpha =\frac{w_{o}-w}{w_{o}-w_{f}}$$
where:*w_o_:* is the weight of the sample at t = 0,*w*: is the weight of the sample at t = t,*w_f_*: is the weight of the sample at the experiment end.

Kinetic triple parameters can be obtained from TGA data using some models derived from Equation (1). The published models use either multiple TGA at different heating rates (called isoconversional or model-free methods) or one single TGA data (called non-isoconversional or model-fitting methods). Kinetic equations of some of the widely used isoconversional and non-isoconversional models (for first-order reactions) are shown in Table 4.

### 2.3. Artificial Neural Network (ANN) Modeling

To model an engineering process, a model must be developed based on available data, and then the model parameters are estimated. However, this is not an easy task especially for complex systems with non-linear relations. Alternatively, the artificial neural network (ANN) modeling can be a promising preferred tool to be used.

An ANN topology has three layers (input, output, and hidden layers of neurons) with different functions. Each layer has a bias vector (weight matrix) and an output vector [15]. Process variables must be fixed initially, and the available data must be representative and fall within the defined variable margin.

Additionally, the ANN architecture consists of several connected layers with their transfer functions. The best network architecture depends on the type of the represented problem. For high performance of ANN-prediction, a genetic algorithm is applied to optimize the ANN parameters such as the number of hidden layers, the number of neurons in each hidden layer, and the momentum and learning rates [14,16].

Recently, some researchers have developed ANN models to predict the pyrolytic behavior of the thermal decompositions of some materials using TGA data [14,17,18,19,20,21,22]. However, in this work, a highly-efficient ANN model is aimed to be developed to predict, for the first time, the catalytic pyrolysis of HDPE. The following statistical parameters are used to evaluate the performance of the developed ANN-model [14,16,23]:(3)$$\mathrm{Average}\;\mathrm{correlation}\;\mathrm{factor}\;\left(R^{2}\right)=1-\frac{\sum {\left({\left(W\;\%\right)}_{est}-{\left(W\;\%\right)}_{\exp}\right)}^{2}}{\sum {\left({\left(W\;\%\right)}_{est}-\bar{{\left(W\;\%\right)}_{\exp}}\right)}^{2}}$$
(4)$$\mathrm{Root}\;\mathrm{mean}\;\mathrm{square}\;\mathrm{error}\;\left(\mathrm{RMSE}\right)=\sqrt{\frac{1}{N}\;\sum {\left({\left(W\;\%\right)}_{\mathrm{est}}-{\left(W\;\%\right)}_{\exp}\right)}^{2}\;}$$
(5)$$\mathrm{Mean}\;\mathrm{absolute}\;\mathrm{error}\;\left(\mathrm{MAE}\right)=\frac{1}{N}\sum \left|{\left(W\;\%\right)}_{est}-{\left(W\;\%\right)}_{\exp}\right|$$
(6)$$\mathrm{Mean}\;\mathrm{bias}\;\mathrm{error}\;\left(\mathrm{MBE}\right)=\frac{1}{N}\sum ({\left(W\;\%\right)}_{est}-\left(W\;\%{)}_{\exp}\right)$$
where *(W %)_est_*, *(W %)_exp_*, and $\bar{\left(W\%\right)}$ are the ANN model-estimated, experimental, and average values of mass left %, respectively.

## 3. Results and Discussion

### 3.1. Kinetics Study of Catalytic Pyrolysis of HDPE

Figure 1, Figure 2 and Figure 3 represent the thermogravimetric analysis (TGA), derivative thermogravimetric (DTG) and conversion curves of the pyrolysis of HDPE at different heating rates and catalyst to polymer ratio, respectively. Both DTG and conversion curves were obtained from the thermogravimetric (TG) data and conversion was calculated using Equation (2). Generally, as the heating rate increases, both DTG and conversion curves are shifted to the right (towards higher temperatures) which implies higher on-set, end-set, and decomposition peak temperatures. On the other hand, both curves are shifted to the left (towards lower temperatures) and polymer starts to degrade at lower temperatures when the catalyst is used due to the polymer catalytic cracking process.

In addition, these figures show clearly that there was only one main reaction region for the catalytic cracking of HDPE which is in a full agreement with the available literature [10]. However, the process of catalytic cracking of HDPE cannot be considered as an elementary reaction, whereas the kinetic parameters derived from TGA are obtained only for a short range of temperatures which represents only the range where the decomposition starts passing through the temperature of the maximum decomposition rate.

Figure 4 clearly shows the effect of the quantity of the used catalyst with respect to the tested polymer at different heating rates (5, 10, and 15 K/min). As the catalyst to polymer ratio increased the conversion increased (as shown in Figure 3) but a lower cracking activity (lower peak temperature) was observed, and the effect of the catalyst mass diminished with increasing the heating rate.

Table 5 summarizes characteristic temperatures (T_onset_, T_5%_, T_peak_, and T_endset_) along with mass loss and residue percentages of the pyrolysis of HDPE at different catalyst to polymer ratios.

Table 6 presents the obtained values of activation energy (*E*) using three isoconversional (Friedman, FWO, and KAS) methods. These tables represent only data for the high conversion range (0.5–0.9) because of the low accuracy of the obtained values at low conversions [24]. In addition, Table 7 represents the activation energy data calculated by two non-isoconversional models (Arrhenius and Coat–Redfern).

The activation energy values, calculated by Friedman, FWO, and KAS at different conversions, shared the same trend as a function of the catalyst to polymer ratio. Additionally, it has been observed that E values calculated by the Friedman model at different catalyst to polymer ratios were the lowest and those obtained by FWO were the highest. However, the KAS model produced intermediate values with the highest R^2^. Different ranges of activation energy values were observed, expected, and reported (see [11,12]) and it depends on the used method. However, as expected by theory, the activation energy of the catalytic thermal decomposition of HDPE is still lower than that of pure HDPE [25,26]. Al-Salem et al. [27] attributed the decrease in the activation energy value to the effect of acid-base of the catalyst. The estimated apparent activation energy for catalytic and thermal cracking of HDPE from different sources fall within the range of 206–445 kJ/mol [27].

However, the average *E* values obtained by two non-isoconversional models were close. In addition, the effect of heating rate was dominating at low catalyst to polymer ratio and *E* values increased as heating increased. However, at high catalyst to polymer ratio, the effect of catalyst was dominating, and E values almost decreased as the ratio increased. Generally, *E* values increase as heating rate increases [28], and using catalysts increases the reaction rate and provides an opportunity for the reaction to occur at lower activation energy [29].

### 3.2. Prediction of Catalyst Pyrolysis by ANN Model

The TGA data of the catalytic pyrolysis of HDPE was targeted to be predicted by a developed highly-efficient ANN model. In the current study, an ANN model with a feed-forward back-propagation neural network (FFBPNN) scheme has been developed to predict the mass left % based on 900 experimental data points. While heating rate, temperature, and HZSM-5/HDPE mass ratio were the input variables, the mass left % was the output parameter of the network. The whole datasets were arbitrarily divided into three sets as follows: 70% (630 datasets) for training, 15% (135 datasets) for validation, and 15% (135 datasets) for testing.

Since the number of input and output layers neurons are fixed, the number of the neurons of the hidden layer is the controlled variable in predicting the performance and the accuracy of the ANN model. Besides, while little number of neurons will lead to the underfitting, which may lead to increase the training error of ANN, too many neurons may cause long time training where new datasets cannot be predicted (overfitting) [30,31].

Table 8 shows the prediction performance of different ANN structures with different numbers of neurons, hidden layers, and transfer functions. The most efficient network structure has been selected based on the value of R^2^. Additionally, mean square error (MSE) has been included as the second criterion.

The ANN topology of the selected model (ANN11), shown in Figure 5, has two hidden layers with 10 neurons and tansig-logsig transfer functions. Although the Levenberg–Marquardt algorithm was used, other algorithms such as the scaled conjugate gradient and Bayesian regularization were tested as well. As shown in Figure 6, a very good agreement between ANN-predicted and experimental results has been observed.

Then, the performance of the selected model was evaluated by different statistical parameters such as R^2^, MAE, RMSE, and MBE. Table 9 lists the values of these parameters. The high value of R^2^ along with very low values of MAE, RMSE, and MBE indicates a high-efficient performance of the selected model [14].

After that, new 45 datasets were tested by the selected model (NN-3-10-10-1) as shown in Table 10 and Figure 7 clearly shows the high performance of the selected network (See Table 11 as well).

## 4. Conclusions

Thermograms of the catalytic pyrolysis of HDPE showed the same shapes and trends at different HZSM-5/HDPE mass ratios and heating rates. Additionally, one reaction region, which can be fitted linearly, was observed and thermal degradation occurred at lower temperatures when the catalyst was used.

In this study, TGA kinetics data was modeled by two methods: using five isoconversional/non-isoconversional models, and a developed highly-efficient ANN model.

In the first method, the activation energy of the catalytic thermal decomposition of HDPE was calculated by Friedman, FWO, KAS, Coats–Redfern, and Arrhenius models at different heating rates and catalyst to polymer ratios.

In the second method, a highly-efficient ANN model, with two hidden layers, and Tansig-Logsig transfer functions, has been developed. Then, new input datasets have been predicted by the proposed ANN structure with a very high value of R^2^ (>0.9998) and very low RMSE, MAE, and MBE. This indicates the capability of the developed model to efficiently predict the non-isothermal TGA data of the catalytic thermal cracking of HDPE.

## Figures and Tables

**Figure 1 polymers-12-01813-f001:**
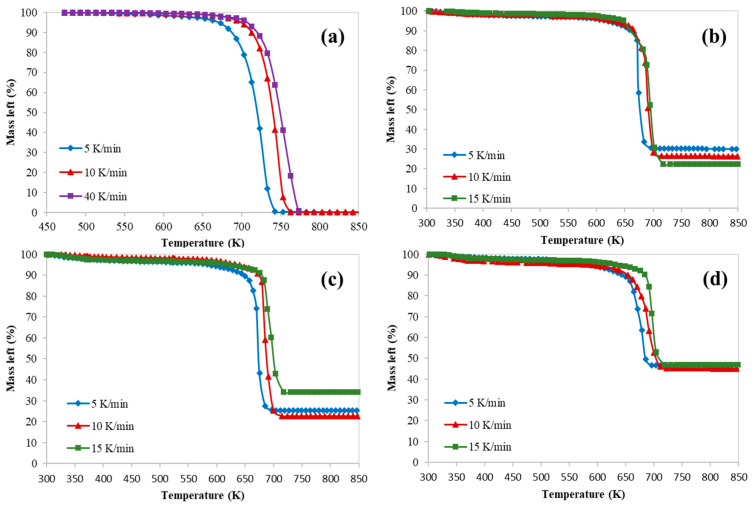
Thermogravimetric analysis (TGA) curves of catalytic pyrolysis of HDPE at different catalyst to polymer mass ratio: (**a**) 0, (**b**) 0.5, (**c**) 0.77, and (**d**) 1.

**Figure 2 polymers-12-01813-f002:**
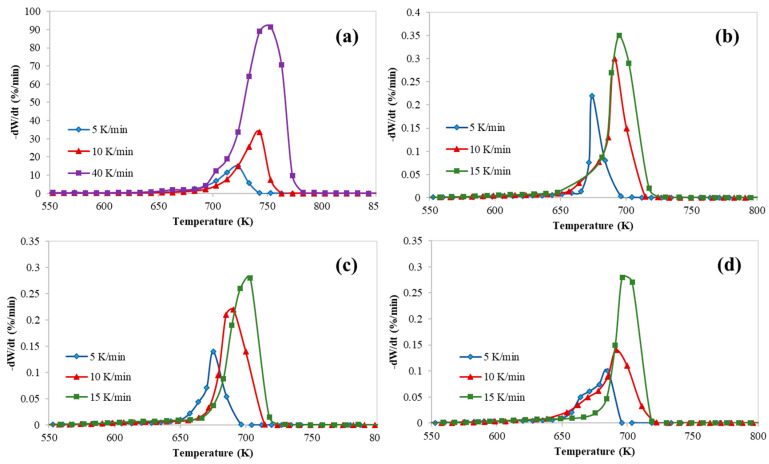
Derivative thermogravimetric (DTG) thermograms of catalytic pyrolysis of HDPE at different catalyst to polymer mass ratio: (**a**) 0, (**b**) 0.5, (**c**) 0.77, and (**d**) 1.

**Figure 3 polymers-12-01813-f003:**
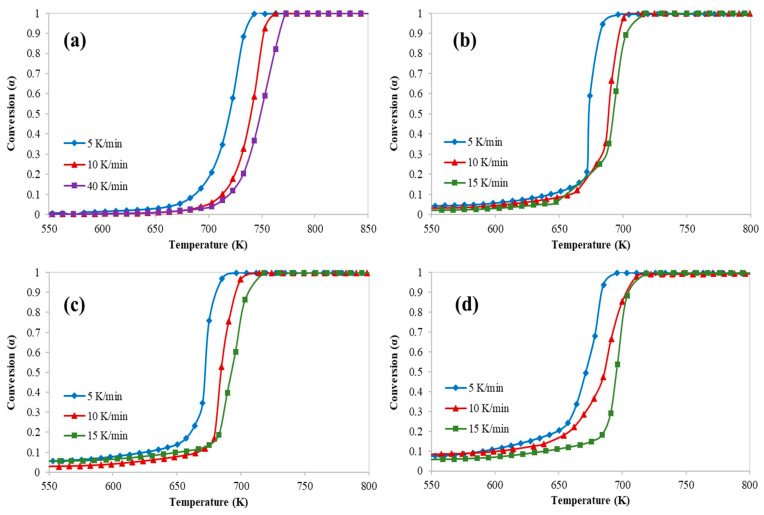
Conversion curves of catalytic pyrolysis of HDPE at different catalyst to polymer mass ratio: (**a**) 0, (**b**) 0.5, (**c**) 0.77, and (**d**) 1.

**Figure 4 polymers-12-01813-f004:**
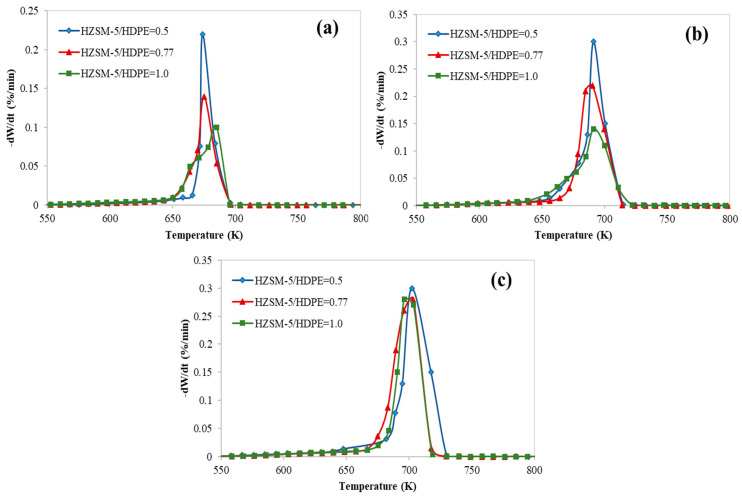
DTG curves at different heating rates: (**a**) 5 K/min, (**b**) 10 K/min, and (**c**) 15 K/min.

**Figure 5 polymers-12-01813-f005:**
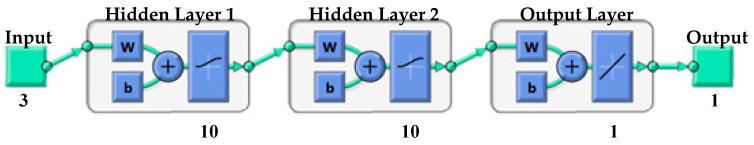
Topology of the selected network.

**Figure 6 polymers-12-01813-f006:**
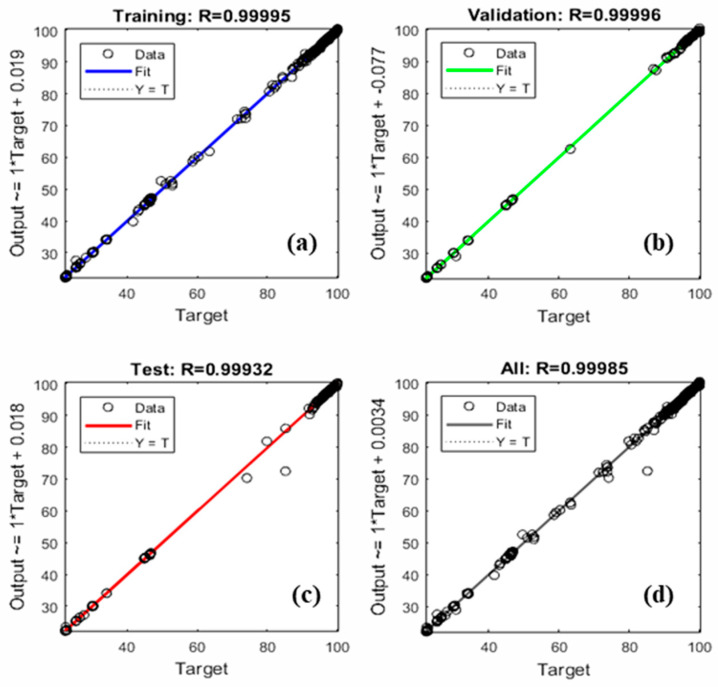
Regression plots of (**a**) training data, (**b**) validation data, (**c**) test data, and (**d**) complete data set of the selected ANN model.

**Figure 7 polymers-12-01813-f007:**
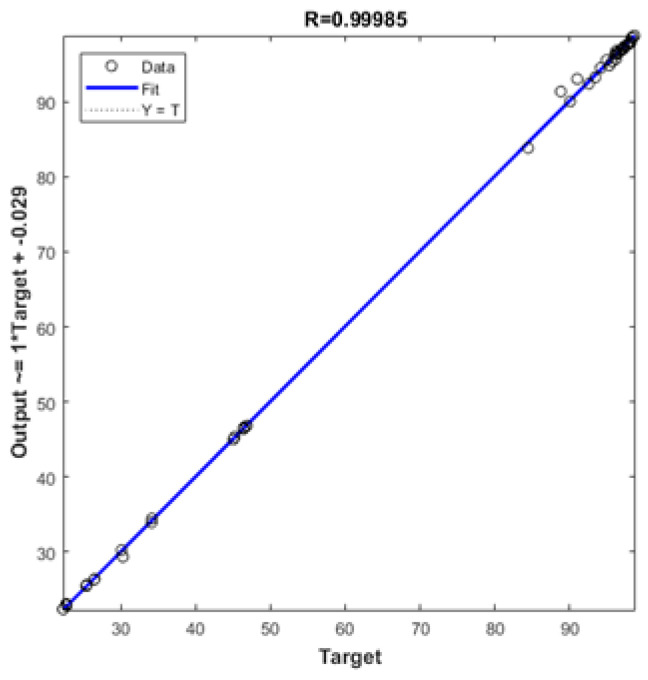
Regression of simulated data for the selected ANN model.

**Table 1 polymers-12-01813-t001:** Ultimate and proximate analysis of high-density polyethylene (HDPE).

Waste Plastic	Proximate Analysis, wt %			Ultimate Analysis, wt %			
	Moisture	Volatile	Ash	C	H	N	S
**HDPE**	4.504	94.278	1.218	78.33	20.71	0.00	0.96

**Table 2 polymers-12-01813-t002:** Specifications of HZSM-5 Catalyst.

Name	SiO_2_/Al_2_O_3_ Mole Ratio	Nominal Cation Form	Na_2_O, wt%	Surface Area, m^2^/g
HZSM-5 (CBV3024E)	30	Hydrogen	0.05	400

**Table 3 polymers-12-01813-t003:** Experimental matrix.

Run No.	1	2	3	4	5	6	7	8	9
**HZSM-5/HDPE** (mass Ratio)	0.5	0.77	1.0	0.5	0.77	1.0	0.5	0.77	1.0
**Heating Rate** (K/min)	5	5	5	10	10	10	15	15	15

**Table 4 polymers-12-01813-t004:** Equations of the selected models [14].

Model	Equation
**Isoconversional**	
Friedman	$\ln\left(\beta \frac{d\alpha }{dT}\right)=\ln\left(A\right)+\ln\left(1-\alpha \right)-\frac{E}{R}\frac{1}{T}$
Flynn–Wall–Qzawa (FWO)	$\ln\left(\beta \right)=\ln\left(-\frac{A\;E}{R\;ln\left(1-\alpha \right)}\right)-5.331-1.052\frac{E}{R}\frac{1}{T}$
Kissinger–Akahira–Sunose (KAS)	$\ln\left(\frac{\beta }{T^{2}}\right)=ln\left(-\frac{A\;R}{E\;ln\left(1-\alpha \right)}\right)-\frac{E}{R}\frac{1}{T}$
**Non-isoconversional**	
Arrhenius	ln(dWdtW)=ln(A)−ER1T
Coats–Redfern	$ln\left[\frac{-ln\left(1-\alpha \right)}{T^{2}}\right]=ln\left[\frac{A\;R}{\beta \;E}\left(1-\frac{2RT}{E}\right)\right]-\frac{E}{R}\frac{1}{T}$

**Table 5 polymers-12-01813-t005:** Thermogravimetric analysis data.

HZSM-5/HDPE (Mass Ratio)	Heating Rate K/min	T_onset_ (K)	T_5%_ (K)	T_peak_ (K)	T_endset_ (K)	Mass Loss (%)	Residue (%)
**0.0** **(Pure HDPE)**	5	473	668	723	743	99	1
	10	473	698	743	763	99	1
	15	473	696	753	773	99	1
**0.5**	5	310	589	674	696	70	30
	10	304	615	691	712	74	26
	15	336	635	695	718	78	22
**0.77**	5	301	512	675	696	75	25
	10	334	622	696	712	77	23
	15	310	505	703	718	66	34
**1.0**	5	325	516	685	696	54	46
	10	316	396	691	713	55	45
	15	311	464	696	718	53	47

**Table 6 polymers-12-01813-t006:** Activation energies calculated by isoconversional models.

Conversion	HZSM-5/HDPE = 0.5		HZSM-5/HDPE = 0.77		HZSM-5/HDPE = 1.0	
	E (kJ/mol)	R^2^	E (kJ/mol)	R^2^	E (kJ/mol)	R^2^
**Friedman Model**						
0.5	120	0.9749	99	0.9949	263	0.9815
0.6	219	0.9811	147	0.9922	211	0.877
0.7	225	0.9977	154	0.9995	141	0.673
0.9	180	0.9552	142	0.9505	100	0.9085
**Average**	**186**	**0.9772**	**135.5**	**0.9843**	**178.7**	**0.8600**
**FWO Model**						
0.5	202	0.9633	196	0.9995	332	0.9997
0.6	191	0.97	181	0.9991	183	1
0.7	198	0.9797	169	0.9997	196	0.9973
0.9	193	0.9824	166	0.9989	168	0.9136
**Average**	**196**	**0.97385**	**178**	**0.9993**	**219.75**	**0.97765**
**KSA Model**						
0.5	201	0.9593	198	0.9787	170	0.9991
0.6	189	0.9665	179	0.999	182	1
0.7	196	0.9774	167	0.9997	195	0.997
0.9	192	0.9803	165	0.9931	191	0.9919
**Average**	**194.5**	**0.970875**	**177.25**	**0.992625**	**184.5**	**0.997**

**Table 7 polymers-12-01813-t007:** Activation energies calculated by non-isoconversional models.

Run	Heating Rate (K/min)	HZSM-5/HDPE Mass Ratio	Coats–Redfern		Arrhenius	
			E (kJ/mol)	R^2^	E (kJ/mol)	R^2^
1	5	0.5	126	0.9399	125	0.9995
4	10	0.5	140	0.9631	142	0.958
7	15	0.5	168	0.9942	168	0.8222
**Average**			**144.7**	**0.9657**	**145**	**0.9266**
2	5	0.77	140	0.9682	157	0.9813
5	10	0.77	145	0.9723	148	0.9103
8	15	0.77	110	0.9414	112	0.9047
**Average**			**131.7**	**0.9606**	**139**	**0.9321**
3	5	1	140	0.9476	129	0.9799
6	10	1	132	0.9997	135	0.9575
9	15	1	123	0.8387	156	0.9423
**Average**			**131.7**	**0.9287**	**140**	**0.9599**

**Table 8 polymers-12-01813-t008:** Prediction performance of different artificial neural network (ANN)-architecture.

Model No.	Network Topology	1st Transfer Function	2nd Transfer Function	R^2^	MSE for Training
ANN1	NN-3-10-1	TANSIG	-	0.99920	1.48
ANN2	NN-3-15-1	TANSIG	-	0.99935	0.904
ANN3	NN-3-5-1	TANSIG	-	0.99722	5.18
ANN4	NN-3-10-1	LOGSIG	-	0.99878	2.64
ANN5	NN-3-15-1	LOGSIG	-	0.99888	1.41
ANN6	NN-3-5-1	LOGSIG	-	0.99824	2.46
ANN7	NN-3-15-15-1	TANSIG	TANSIG	0.99981	0.0635
ANN8	NN-3-15-15-1	LOGSIG	TANSIG	0.99962	0.0646
ANN9	NN-3-15-15-1	TANSIG	LOGSIG	0.99972	0.0571
ANN10	NN-3-10-15-1	TANSIG	LOGSIG	0.99983	0.0422
**ANN11**	**NN-3-10-10-1**	**TANSIG**	**LOGSIG**	**0.99985**	**0.0860**
ANN12	NN-3-10-10-1	LOGSIG	LOGSIG	0.99970	0.227
ANN13	NN-3-15-15-1	LOGSIG	LOGSIG	0.99904	1.08
ANN14	NN-3-10-15-1	LOGSIG	LOGSIG	0.99960	0.176
ANN15	NN-3-15-10-1	LOGSIG	LOGSIG	0.99970	0.0504
ANN16	NN-3-20-20-1	LOGSIG	LOGSIG	0.99974	0.0479
ANN17	NN-3-20-20-1	TANSIG	LOGSIG	0.99956	0.274
ANN18	NN-3-9-9-1	TANSIG	LOGSIG	0.99884	1.28
ANN19	NN-3-11-11-1	TANSIG	LOGSIG	0.99968	0.276

**Table 9 polymers-12-01813-t009:** Statistical parameters of the ANN11 model.

Set	Statistical Parameters			
	R^2^	RMSE	MAE	MBE
Training	0.99995	0.30721	0.16388	0.01954
Validation	0.99996	0.27855	0.17212	−0.00186
Test	0.99932	1.19340	0.30865	−0.15220
**All**	**0.99985**	**0.53975**	**0.18683**	**−0.00943**

**Table 10 polymers-12-01813-t010:** ANN-predicted results of the new input data.

No.	Input Data			Output Data
	Heating Rate (K/min)	Temperature (K)	HZSM-5/HDPE (Mass Ratio)	Mass Left (%)
1	5	402.59	0.5	98.03
2	5	502.6	0.5	97.32
3	5	605.73	0.5	95.68
4	5	704.33	0.5	30.28
5	5	801.36	0.5	30.09
6	5	411.03	0.75	97.04
7	5	510.32	0.75	96.29
8	5	612.96	0.75	93.43
9	5	711.87	0.75	25.45
10	5	809.3	0.75	25.36
11	5	419.81	1	98.02
12	5	523.09	1	96.26
13	5	620.13	1	92.56
14	5	726.16	1	46.45
15	5	823.51	1	46.39
16	10	433.07	0.5	98.32
17	10	533.25	0.5	97.67
18	10	631.75	0.5	94.95
19	10	733	0.5	26.49
20	10	832.17	0.5	26.44
21	10	441.34	0.75	98.4
22	10	541.26	0.75	97.87
23	10	648.03	0.75	94.17
24	10	740.81	0.75	22.74
25	10	847.67	0.75	22.69
26	10	457.84	1	96.09
27	10	557.59	1	95.32
28	10	653.87	1	90.08
29	10	755.36	1	45.2
30	10	853.97	1	45.03
31	15	469.55	0.5	98.66
32	15	566.83	0.5	98.17
33	15	665.32	0.5	88.83
34	15	768.01	0.5	22.29
35	15	867.59	0.5	22.26
36	15	479.36	0.75	96.78
37	15	576.52	0.75	96.22
38	15	675.12	0.75	91.02
39	15	776.31	0.75	34.14
40	15	875.47	0.75	34.11
41	15	488.8	1	97.24
42	15	585.83	1	96.59
43	15	690.8	1	84.45
44	15	785.3	1	46.84
45	15	884.5	1	46.79

**Table 11 polymers-12-01813-t011:** Statistical parameters of the ANN12 model for the newly tested data.

Set	Statistical Parameters			
	R^2^	RMSE	MAE	MBE
New Tested Data	0.99984	0.55767	0.29725	0.08688

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
