# Peer review of "Application of Artificial Neural Networks to Predict the Catalytic Pyrolysis of HDPE Using Non-Isothermal TGA Data"

_polymers, 2020, doi:10.3390/polym12081813_

Round 1

Reviewer 1 Report

A first reading of the paper entitled as “Application of Artificial Neural Networks to Predict the Catalytic Pyrolysis of HDPE using Non-Isothermal TGA Data” appears as well written and interesting.

The relative novelty of the paper lies in the use of Artificial Neural Network (ANN) models to manage and predict the kinetic parameters; equally than the authors did in a recent article based on recycled LDPE. Since the authors properly identify a polymer as a complex system, this reviewer agrees completely with this approach. However, the authors do not provide enough information about the polymer used (and so null information about its complexity), they simply indicate the following for the polymer (HDPE Ipoh SY Recycle Plastic, Perak, Malaysia), without information about molecular weight, distribution, thermal properties, other properties, et cetera. If the absence of these data, the article becomes impossible to be replicated for whoever person interested. Note that this is the essence of Science. So, for traceability reasons provide the required information. In the case that the supplier does not have this information, I´m afraid that this task must be performed in the authors' labs. Since Polymers is a polymer science related journal, more details of the polymer used than just the name (not even the grade is provided) must be given by the authors. As mentioned before, if the absence of these, the article is not traceable and so it cannot be admitted in a scientific journal.

Please, note that in the absence of the latter your article passes to be just a mere case study of an unknown material instead of being a superb article in light of the mathematical approach performed. Please provide this information.

The rest of the article is Ok for this reviewer, but however I would like to recommend comparing it with other HDPE in the market in order to ascertain the influence of the polydispersity, the additives, the molecular weight, and so on, on the final predictions by using ANN models. Nevertheless, this is just a non-mandatory suggestion but this reviewer thinks that a comparison with other HDPE grades rather than with just other limited mathematical models would provide extra robustness to the study, and would provide a stronger idea about the advantages of ANN.

To mention that I have enjoyed very much of the article, and so, I would like to recommend the publication of it if at least the concerns about the properties of the HDPE used are solved.

The queries pointed out by this reviewer are very easy to be solved but since they are focused in the experimental section, and according to Polymers instructions about, the recommendation must be MAJOR REVISION.

Author Response

Response to the Comments of Reviewer 1

Comments and Suggestions for Authors

A first reading of the paper entitled as “Application of Artificial Neural Networks to Predict the Catalytic Pyrolysis of HDPE using Non-Isothermal TGA Data” appears as well written and interesting.

The relative novelty of the paper lies in the use of Artificial Neural Network (ANN) models to manage and predict the kinetic parameters; equally than the authors did in a recent article based on recycled LDPE. Since the authors properly identify a polymer as a complex system, this reviewer agrees completely with this approach.

Thanks

However, the authors do not provide enough information about the polymer used (and so null information about its complexity), they simply indicate the following for the polymer (HDPE Ipoh SY Recycle Plastic, Perak, Malaysia), without information about molecular weight, distribution, thermal properties, other properties, et cetera. If the absence of these data, the article becomes impossible to be replicated for whoever person interested. Note that this is the essence of Science. So, for traceability reasons provide the required information. In the case that the supplier does not have this information, I´m afraid that this task must be performed in the authors' labs. Since Polymers is a polymer science related journal, more details of the polymer used than just the name (not even the grade is provided) must be given by the authors. As mentioned before, if the absence of these, the article is not traceable and so it cannot be admitted in a scientific journal. Please, note that in the absence of the latter your article passes to be just a mere case study of an unknown material instead of being a superb article in light of the mathematical approach performed. Please provide this information.

Thanks for your kind valuable comment. Please be informed that the available physical properties of the used HPDE presented in Table 1, have been added recently to the manuscript.

Table 1. Physical properties of high-density polyethylene.

Manufacturer

Ipoh SY Recycle Plastic, Perak, Malaysia

Polymer Type

Polymer Type

Recycled HDPE

HDPE N2

(pelletization at nitrogen environment)

Appearance (at 25 °C)

Solid

Physical State

Pellets

Colour

Light Brown

Density (Kg/m3)

940-960

Melting Temperature (°C)

~134

The rest of the article is Ok for this reviewer, but however I would like to recommend comparing it with other HDPE in the market in order to ascertain the influence of the polydispersity, the additives, the molecular weight, and so on, on the final predictions by using ANN models. Nevertheless, this is just a non-mandatory suggestion but this reviewer thinks that a comparison with other HDPE grades rather than with just other limited mathematical models would provide extra robustness to the study, and would provide a stronger idea about the advantages of ANN.

It is an appreciated comment.

However, upon the authors knowledge, developing a highly efficient ANN model to predict the catalytic pyrolysis of HDPE has been performed for the first time and there is no data to be compared with.

To mention that I have enjoyed very much of the article, and so, I would like to recommend the publication of it if at least the concerns about the properties of the HDPE used are solved.

Thanks

The queries pointed out by this reviewer are very easy to be solved but since they are focused in the experimental section, and according to Polymers instructions about, the recommendation must be MAJOR REVISION.

Thanks with best regards,

Dr. Ibrahim Dubdub

Reviewer 2 Report

The paper entitled: "Application of Artificial Neural Networks to Predict the Catalytic Pyrolysis of HDPE using Non-Isothermal TGA Data", by Mohammed Al-Yaari and Ibrahim Dubdub presents a comprehensive kinetic study of the catalytic pyrolysis of high density polyethylene (HDPE) utilizing thermogravimetric analysis (TGA) data. Thermograms of the catalytic pyrolysis of HDPE were analysed and kinetic data calculated by Friedman, FWO, KAS, Coats- Redfern, and Arrhenius models at different heating rates and catalyst to polymer ratios. The results are clear, the conclusions are not in doubt, the work is well written. The literature is cited accordingly.

I recommend the manuscript for publication as is.

Author Response

Response to the Comments of Reviewer 2

Comments and Suggestions for Authors

The paper entitled: "Application of Artificial Neural Networks to Predict the Catalytic Pyrolysis of HDPE using Non-Isothermal TGA Data", by Mohammed Al-Yaari and Ibrahim Dubdub presents a comprehensive kinetic study of the catalytic pyrolysis of high density polyethylene (HDPE) utilizing thermogravimetric analysis (TGA) data. Thermograms of the catalytic pyrolysis of HDPE were analysed and kinetic data calculated by Friedman, FWO, KAS, Coats- Redfern, and Arrhenius models at different heating rates and catalyst to polymer ratios. The results are clear, the conclusions are not in doubt, the work is well written. The literature is cited accordingly.

I recommend the manuscript for publication as is.

Many thanks.

Best regards,

Dr. Ibrahim Dubdub

Round 2

Reviewer 1 Report

See attached file

Author Response

Response to the Comments of Reviewer 1

Dear Respected Reviewer,

Greetings

We appreciate the time and effort that you dedicated to providing feedback on our manuscript and we are grateful for the insightful comments on and valuable improvements to our paper.

Although the requested tests are helpful and easy to be conducted at normal conditions, they cannot be done during the current coronavirus pandemic. As acknowledged in the 1st version of the manuscript, please be informed that experimentation was conducted at the University of Science, Malaysia (USM) and unfortunately we do not have samples of the used polymer in our labs (in Saudi Arabia) to be tested. We have tried to ask some of our colleagues at USM to help us in this regard, but they could not because of their limited mobility nowadays and lockdown conditions.

In addition, I would like to highlight the followings comments/findings which will help the target audience of the Polymers journal:

  1. As stated clearly by the manuscript title, the main objective of this work is to develop a highly efficient ANN model to predict the catalytic pyrolysis of HDPE which you have thankfully acknowledged that in your 1st kind feedback. Upon the authors' knowledge, this is the first work conducted in this regard and thus it will attract the kind attention of interested researchers with no doubt. Also, the 3rd input parameter (catalyst to polymer ratio) to the proposed ANN network has not been adopted, even for other polymer systems, before this work. Furthermore, step by step modeling protocol has been reported to be easily followed. This useful protocol is independent of the polymer type and characteristics.
  2.  In general, ANN models are used for forecasting new data to save the researchers time, money, and effort. As you know, some experimental data may cost us a lot to be collected and in many cases some experiments cannot be conducted due to limited resources/facilities, safety, and economic concerns.
  1. As the ultimate goal of all researchers, interested in polymer recycling, is the environmental sustainability, plastic wastes are targeted to be reduced. Therefore, recycled HDPE, from a well-known Malaysian company, was used for our experimentations.
  1. In addition, the following investigations can help interested researchers as well:
    1. Robustness of three well-known isoconversional models and two non-isoconversional models to obtain some kinetic parameters of a recycled HDPE catalytic pyrolysis.
    2. TGA, DTG, and conversion curves along with thermograms properties of the catalytic thermal degradation of HDPE.
    3. Effect of heating rate, and catalyst to polymer ratio on the HDPE catalytic pyrolysis.
    4.  When our results of the thermal analysis were compared with the reviewed available published data, no contradictions were observed. Therefore, there was no need for extra tests or adding extra parameters to the designed experimental matrix to explore the reasons behind that.

Your kind understanding is highly appreciated.

Thanks with best regards,

Dr. Ibrahim Dubdub

Round 3

Reviewer 1 Report

Just to say, that this reviewer agrees with all the suggested by the authors about the importance of ANN and all the other concerns of their work. In fact, this has been always my opinion about your study. So, this reviewer does not understand why a seemingly superb article is unable to ensure the traceability of its results by providing complete information about the polymer used.

I may understand the reasons of the authors about the COVID-19 pandemic derived difficulties, but the mandatory information asked by this reviewer is so basic that a simple e-mail from the Malaysia University colleagues can solve this concern. In fact, this is a problem that almost all of us are suffering, and however it doesn´t means that our exigency in our scientific work quality must decay (and much more if this is traduced in an incomplete scholar publication).

Nevertheless, it is hard to accept that you have not a minimum repository of material as to perform a simple FTIR analysis (this is simply the at least mandatory information what this reviewer asked for) since the stock of samples is a critical step in whatever traceability chain. 

In any case, I suggest to the authors to ask the editorial office for more time till the authors are able to provide the requested information in the previous revision draft.

I´m sorry, but in the light of the expressed by the authors, the opinion of this reviewer must remain as to perform the requested Major Revision.

Author Response

Response to the Comments of Reviewer 1

Comments and Suggestions for Authors

Just to say, that this reviewer agrees with all the suggested by the authors about the importance of ANN and all the other concerns of their work. In fact, this has been always my opinion about your study. So, this reviewer does not understand why a seemingly superb article is unable to ensure the traceability of its results by providing complete information about the polymer used.

I may understand the reasons of the authors about the COVID-19 pandemic derived difficulties, but the mandatory information asked by this reviewer is so basic that a simple e-mail from the Malaysia University colleagues can solve this concern. In fact, this is a problem that almost all of us are suffering, and however it doesn´t means that our exigency in our scientific work quality must decay (and much more if this is traduced in an incomplete scholar publication).

Nevertheless, it is hard to accept that you have not a minimum repository of material as to perform a simple FTIR analysis (this is simply the at least mandatory information what this reviewer asked for) since the stock of samples is a critical step in whatever traceability chain. 

In any case, I suggest to the authors to ask the editorial office for more time till the authors are able to provide the requested information in the previous revision draft.

I´m sorry, but in the light of the expressed by the authors, the opinion of this reviewer must remain as to perform the requested Major Revision.

Firstly, it’s our pleasure to express our sincere thanks for your kind valuable and supported advice. In addition, please be informed that we have received the characterization data of the used polymer (HDPE), as requested, from University of Science, Malaysia. The following text and table have been added to the revised manuscript to ensure the reproducibility of our results and findings.

Line 70-73:

The proximate and ultimate analysis of HDPE were conducted using PerkinElmer Simultaneous Thermal Analyzer (STA) 6000, and PerkinElmer 2400 Series II CHNS Elemental Analyzer, USA, respectively. The characterization data are presented in Table 1.

Line 85-86:

Table 1. Ultimate and proximate analysis of HDPE.

Waste Plastic

Proximate Analysis, wt%

Ultimate Analysis, wt%

Moisture

Volatile

Ash

C

H

N

S

HDPE

4.504

94.278

1.218

78.33

20.71

0.00

0.96

Thanks again with best regards,

Dr. Ibrahim Dubdub

Round 4

Reviewer 1 Report

The authors provide the elemental analysis of the polymers used. Although it is not the best way to characterize a macromolecular system as a polymer is, in attention to the superb article, this reviewer may consider that as enough since the authors use a recycled based polymer system.

Nevertheless, a estimation about the molecular weigth and the molecular weigth distribution is more appropiate for a polymer than any other characterization technique.

Another weak point in the article is the absence of information about the degree of previous degradation of the material since this is a recycled one.

Nevertheless, the work is so interesting for the use of the neural networks for predicting the behaviour of the material that this reviewer want to recomend the publication of the article in the actual state.

Additionally, if possibly, this reviewer would like to recommend the use of pristine polymers instead of recycled grades in future works usinf neural networks to predict the pyrolisis in order to make idea about the real possibilities of prediction of the method used. 

Author Response

Response to the Comments of Reviewer 1

Comments and Suggestions for Authors

The authors provide the elemental analysis of the polymers used. Although it is not the best way to characterize a macromolecular system as a polymer is, in attention to the superb article, this reviewer may consider that as enough since the authors use a recycled based polymer system.

Many thanks for your kind understanding.

Nevertheless, an estimation about the molecular weight and the molecular weight distribution is more appropriate for a polymer than any other characterization technique.

Although molecular weight and molecular weight distribution can be used as the polymer fingerprint, proximate and ultimate analyses are more informative and urgently needed for polymer pyrolysis. Proximate and ultimate analyses can give information about feedstock suitability for pyrolysis process and the pyrolysis product quality, respectively.

Another weak point in the article is the absence of information about the degree of previous degradation of the material since this is a recycled one.

We do believe that it depends on the research scope.

Nevertheless, the work is so interesting for the use of the neural networks for predicting the behaviour of the material that this reviewer wants to recommend the publication of the article in the actual state.

Many thanks

Additionally, if possibly, this reviewer would like to recommend the use of pristine polymers instead of recycled grades in future works using neural networks to predict the pyrolysis in order to make idea about the real possibilities of prediction of the method used.

Thanks for your kind recommendation for future work.

Thanks again with best regards,

Dr. Ibrahim Dubdub
